# Neutral beam microscopy with a reciprocal space approach using magnetic beam spin encoding

Morgan Lowe [1], Yosef Alkoby[1], Helen Chadwick [1] & Gil Alexandrowicz [1] ✉

The emerging technique of neutral beam microscopy offers a non-perturbative way of imaging surfaces of various materials which cannot be studied using conventional microscopes. Current neutral beam microscopes use either diffractive focusing or pin-hole scanning to achieve spatial resolution, and are characterised by a strong dependence of the imaging time on the required resolution. In this work we introduce an alternative method for achieving spatial resolution with neutral atom beams which is based on manipulating the magnetic moments of the beam particles in a gradient field, and is characterised by a much weaker dependence of the imaging time on the image resolution. The validity of the imaging approach is demonstrated experimentally by reconstructing one dimensional profiles of the beam which are in good agreement with numerical simulation calculations. Numerical simulations are used to demonstrate the dependence of the signal to noise on the scan resolution and the topography of the sample, and assess the broadening effect due to the spread of velocities of the beam particles. The route towards implementing magnetic encoding in high resolution microscopes is discussed.

Over the last four decades significant achievements have been made towards developing scanning helium microscopes (SHeM)[1], which use a neutral beam of very low energy (sub-eV) helium atoms to image the surface. The inert and neutral properties of helium make SHeM suitable for imaging surfaces that are either damaged, altered, or simply difficult to image using electron and ion microscopy techniques, as well as samples that are transparent or photo-sensitive.

While the short deBroglie wavelength of thermal helium beams results in sensitivity to sub-nanometre topographical features[2], obtaining lateral resolution with a neutral beam of atoms is a significant challenge. Different technologies were developed to achieve lateral resolution, including focusing of neutral atoms using bent single crystals[3,4], quantum reflection from a quartz surface[5], and diffraction from Fresnel zone plates[6], with the latter method leading to the first complete 2d images of a transmission mask[7]. More recent SHeM designs have focused on the conceptually simpler pin-hole technology, which proved to be particularly effective and practical[8–10]. A

common aspect of all the approaches used to date is their reliance on small apertures to achieve a high spatial resolution, whether it is micrometre-sized beam sources in instruments based on focusing elements or a combination of several apertures needed in pin-hole technology. The loss of signal associated with micrometre-sized apertures often results in compromising on the resolution for the sake of obtaining high enough signal-to-noise ratios within acceptable acquisition times.

In this paper, we present an alternative method to obtain spatial resolution with neutral particle beam imaging. The approach is based on magnetically manipulating the magnetic moment of the neutral particles to encode the spatial position of the beam particles before or after interacting with the sample. The technique is an atomic beam analogue of phase encoding Fourier imaging in magnetic resonance imaging (MRI) measurements[11,12]. The image is obtained by performing a series of measurements in reciprocal space, with all the beam particles contributing to the signal in every measurement, which can

[1]Department of Chemistry, Faculty of Science and Engineering, Swansea University, Swansea SA2 8PP, UK. ✉e-mail: g.n.alexandrowicz@swansea.ac.uk

dramatically reduce the scaling of the measurement time with the spatial resolution in comparison to pin-hole imaging. Below, we describe the imaging principle, present proof of the principle of one-dimensional (1d) transmission experiments accompanied by numerical simulations and finally discuss the future prospects of the magnetic-encoding approach for neutral beam microscopy.

## Results

### The basic principles of magnetic encoding beam microscopy

The imaging technique presented below uses the response of the magnetic moments of the beam particles to external fields, as a way of encoding their positions within the plane perpendicular to the beam axis. As a first demonstration of the methodology, we chose a beam of $^3$He atoms. Low energy $^3$He beams are similar to the $^4$He beams used in SHeM applications in terms of surface sensitivity and inertness while having a non-zero magnetic moment which is essential for magnetic encoding imaging. The magnetic moment in the case of $^3$He arises from the nuclear spin, $I = 1/2$. The simplicity of this two-level system allows us to describe the magnetic moment classically[13].

We start with a very simplistic description of a 1d imaging scheme. A particle beam, propagating along the $\hat{\mathbf{z}}$-axis is passed through a beam polariser, after which we can consider the magnetic moment of all the particles to be oriented along the positive $\hat{\mathbf{x}}$-axis. Figure 1 illustrates what happens next to the magnetic moments of the atoms (arrows encapsulated within the circles). The beam interacts with a sample that we want to image. The density distribution function, $\rho(x,y)$ of the continuing beam, which could be moving straight forward in a transmission experiment or reflected back at some angle in a scattering experiment, is a product of the initial density distribution of the beam $D(x,y)$ and a second function $P(x,y)$ which describes either the probability of passing the target in a transmission experiment or scattering into the detector angle in a scattering experiment. It is $P(x,y)$ which contains the information about the sample and how it interacts with the probe, and is what we would like to determine in a microscopy experiment. In the simplistic transmission example shown in Fig. 1, $P(x,y)$ is 0 for $x$ positions that hit the sample and 1 for all other coordinates the beam occupies. Next, the atoms enter an encoding device which creates a magnetic field oriented along the $\hat{\mathbf{y}}$-axis. The field has an overall length $L$ along the beam propagation direction and an amplitude that varies linearly

as a function of the $x$ coordinate of the beam particles, i.e. $\mathbf{B}_{\text{encoding}} = (0, \frac{dB_y}{dx}x, 0)$. Within $\mathbf{B}_{\text{encoding}}$, which acts like a phase gradient in magnetic resonance imaging[11,12] the magnetic moments precess within the $xz$ plane at a Larmor frequency, which depends on the strength of the field at their specific $x$ position[11]. A $^3$He atom moving with velocity $\mathbf{v}$, displaced by $x$ from the centre of the beamline, will, at the end of the encoding field, accumulate a total classical spin phase with respect to the $\hat{\mathbf{x}}$-axis given by the product of the Larmor frequency, $\omega_L = \gamma|\mathbf{B}_{\text{encoding}}|$ and the time it spends in the field $t = \frac{L}{|\mathbf{v}|}$, i.e. $\phi = \gamma\frac{dB_y}{dx}x\frac{L}{|\mathbf{v}|}$ where $\gamma$ is the gyromagnetic ratio of $^3$He.

The beam then continues through a spin analyser, which transmits particles toward a particle detector with a probability related to the projection of their magnetic moment on a specific axis we denote the analyser projection axis.

To unambiguously encode the position of the particles in the beam, we will need to perform two types of measurements: one for an analyser projection axis oriented along the same axis the beam was polarised along ($S_0$) and a second for an analyser projection axis which is orthogonal ($S_{90}$). The two types of measurements can be combined as the real and imaginary components of a complex signal $S(k_x) = S_0 + iS_{90}$, and as we show in the supplementary information (S7), the complex signal can be written as

$$S(k_x) \propto \iint_{-\infty}^{\infty} \rho(x,y)e^{2\pi i k_x x}dxdy + C = \int_{-\infty}^{\infty} \rho_{1D}(x)e^{2\pi i k_x x}dx + C \quad (1)$$

where $k_x = \frac{1}{2\pi}\gamma\frac{L}{|\bar{\mathbf{v}}|}\frac{dB_y}{dx}$, is an experimentally controlled variable calculated for the average velocity of the beam, $|\bar{\mathbf{v}}|$, and $\rho_{1D}(x) = \int \rho(x,y)dy$ is a 1d projection of the density distribution function $\rho(x,y)$ onto the $\hat{\mathbf{x}}$-axis, i.e. the profile of the beam after interacting with the sample, along that axis. The last term $C$ is a constant that can be subtracted from the measurement and will be disregarded in the discussion below.

Equation (1) shows that the complex signal measured in the simple experimental scheme described above is a Fourier transform of the beam profile. Consequently, if the complex signal is measured for a range of gradient field values (between $k_{x\text{min}}$ and $k_{x\text{max}}$), applying an inverse Fourier transformation to the result will reconstruct the profile of the beam, $\rho_{1D}(x) \propto \int_{k_{x\text{min}}}^{k_{x\text{max}}} S(k_x)e^{-2\pi i k_x x}dk_x$. Supplementary Movie 1, illustrates graphically the process of measuring $S(k_x)$, followed by a Fourier transform to reconstruct the profile of the beam and object which partially blocked it. We would like to emphasise that while magnetic encoding in a transmission experiment can be performed before or after the interaction, for a scattering experiment, encoding before the interaction has the advantage of avoiding complications related to the angular spread of the scattered beam.

For most applications, a two-dimensional (2d) image is more useful than a 1d profile. Such an image can be obtained by adding a second encoding field along the beam line, which produces a gradient along the y coordinate. This second gradient can be described by a second reciprocal space variable $k_y = \frac{1}{2\pi}\gamma\frac{L}{|\bar{\mathbf{v}}|}\frac{dB_y}{dy}$ (It should be noted that there is freedom in choosing the direction of the 2nd encoding field; the important property is the direction of the field gradient, which selects the encoding axis). The total spin phase accumulated in this configuration is a simple sum of the phase accumulation in each encoding device, and as shown in the supplementary information (S7), the 2d complex signal, which is measured by scanning both $k_x$ and $k_y$ is a 2d Fourier transform of $\rho(x,y)$, i.e.

$$S(k_x, k_y) \propto \iint \rho(x,y)e^{2\pi i[k_x x + k_y y]}dxdy + C \quad (2)$$

In this case $\rho(x,y)$ can be reconstructed by measuring signals for a sufficiently large range of $k_x$, $k_y$ values and inverse transforming the 2d

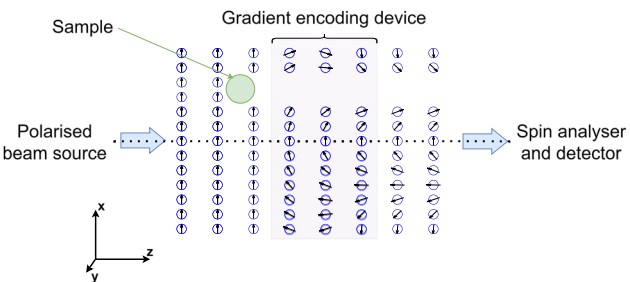

**Fig. 1 | Simplified 1d encoding scheme in a transmission mode experiment.** The beam particles (blue circles) move from left to right along the $\hat{\mathbf{z}}$-axis, and the arrows encapsulated within the circles represent the magnetic moment of the beam particles. The polarised beam has all its magnetic moments oriented along $\hat{\mathbf{x}}$ initially. The beam interacts with a sample, which blocks some of the trajectories, resulting in $P(x,y)$ being zero for specific $x$, $y$ positions. As the particles move within the encoding device, their magnetic moments precess in the $xz$ plane in a way that depends on the field strength and correspondingly on their $x$ position. The precession stops when the particles leave the encoding device. A spin analyser transmits the particles to a detector in a way that depends on the orientation of their magnetic moment with respect to the analyser projection axis. Note that the missing trajectories which were blocked by the sample change the average magnetic moment orientation of the beam and correspondingly, the signal the detector will measure.

signal matrix $S(k_x, k_y)$, in direct analogy to a single slice MRI phase gradient scan[11].

## The experimental setup used to demonstrate the imaging principle

To experimentally realise the phase encoding method described above, we used the magnetic molecular interferometer (MMI) setup[14–16]. The MMI was designed to perform rotationally controlled molecule-surface scattering experiments. However, it also has a straight-through arm[17], which we modified slightly to perform 1d imaging experiments in transmission mode. Due to various constraints, which will be explained below, the experimental setup is more complicated than the simplified imaging scheme described earlier and illustrated above in Fig. 1 and supplementary Movie 1. The setup includes several additional magnetic fields with non-intuitive orientations, as well as a different scheme for measuring the two components of the complex signal.

The setup, which is drawn schematically in Fig. 2, is briefly described below and further technical details can be found in the methods section (Note that for clarity, the description below follows the vector labelling convention introduced above differing from previous conventions describing the MMI[14–16]).

A continuous beam of low-energy $^3$He atoms is created by supersonically expanding gas through a cold (40 K) nozzle into the vacuum and passing the beam through a skimmer[18]. A hexapole magnet followed by a dipole field is used to produce a parallel and polarised beam[19,20]. At the exit of this element, we can consider all the magnetic moments to be oriented along $\hat{x}$. The beam then enters a solenoid producing a controllable homogeneous magnetic field, $\mathbf{B}_1$, oriented along the $\hat{z}$-axis, the role of which will become apparent later. A mechanical linear translator is used to position a 100 μm diameter wire (aligned along the $\hat{y}$-axis) into the beam path at a well-defined $x$ position. The wire was used to both modify the spatial distribution of the imaged beam by blocking it at particular positions, and also to perform independent reference measurements of the beam profile described later. The continuing beam enters the encoding gradient device which includes two magnetic fields ($\mathbf{B}_{encoding}$ and $\mathbf{B}_2$) for reasons which are explained below. A second combination of dipole and hexapole fields are used as a spin analyser[20,21] passing particles to the detector in a way which is proportional to the projection of their magnetic moment onto the $\hat{x}$-axis. The detected signal is the partial pressure of $^3$He measured by a mass spectrometer (Hiden HAL-201).

To understand the need for $\mathbf{B}_1$ and $\mathbf{B}_2$, which are not part of the simple scheme illustrated in Fig. 1, we need to first describe the encoding field. To achieve a magnetic field gradient, we used a 12-wire configuration carrying currents that follow a pattern of $I = I_0 \cos(2\theta)$[11], details of which are given in Supplementary Fig. S4 (with an assessment of field quality given in S1). A common problem that arises when creating gradient fields is that in addition to the gradient we are interested in ($\frac{dB_y}{dx}$ in our case), they produce other gradients within the region of interest. As can be seen in the inset plot in Fig. S4, for the 12-wire configuration, this is a $\frac{dB_x}{dy}$ component that could spoil the encoding scheme. Our solution was to superimpose a second much stronger homogenous field, $\mathbf{B}_2$, along the $\hat{y}$ direction. Because of the orthogonality of the strong field to the unwanted $B_x$ component, the effect of the latter on the spin precession can be reduced dramatically.

Whilst the strong homogenous field, $\mathbf{B}_2$, solves the problem of the unwanted orthogonal gradient, it also introduces a complication. Particle beams will always have some spread of velocities, typically a full-width half maximum (FWHM) of a few percent in supersonic beams[18], leading to slightly different flight times and phase accumulation within a magnetic field. Since $\mathbf{B}_2$ is a relatively strong magnetic field (we used 170 G), it will lead to a complete dephasing of the magnetic moment of particles with different velocities and a loss of signal. Fortunately, this effect can be reversed by passing the beam through another homogeneous magnetic field with a field integral magnitude identical to that produced by $\mathbf{B}_2$, leading to a refocusing condition known as a spin echo signal[13,22–24]. The refocusing field $\mathbf{B}_1$ was implemented with a solenoid electromagnet which is an integral part of the MMI apparatus[14] and can be controlled with ppm accuracy. It is important to note that $\mathbf{B}_1$ points along the $\hat{z}$-axis, whereas both the encoding field and the homogeneous field, point along the $\hat{y}$-axis. While this makes it harder to intuitively understand how the velocity-dependent dephasing is reversed, the echo that it produces can be understood within the multiple echo picture, which takes into account different orientations of precession fields[25]. We have also included an animation to illustrate this graphically in Supplementary Movie 2.

Finally, we would like to point to another difference between the simplistic encoding scheme described above and the setup we used. The orientation of both the polariser and analyser in the MMI setup is fixed along the $\hat{x}$ axis and cannot be easily changed. Our solution to measuring both the real and imaginary components was to rotate the magnetic moments of the incident beam as it passes through the additional homogenous field $\mathbf{B}_1$ mentioned above. Identifying the currents which correspond to 0° and 90° rotations allows us to mimic

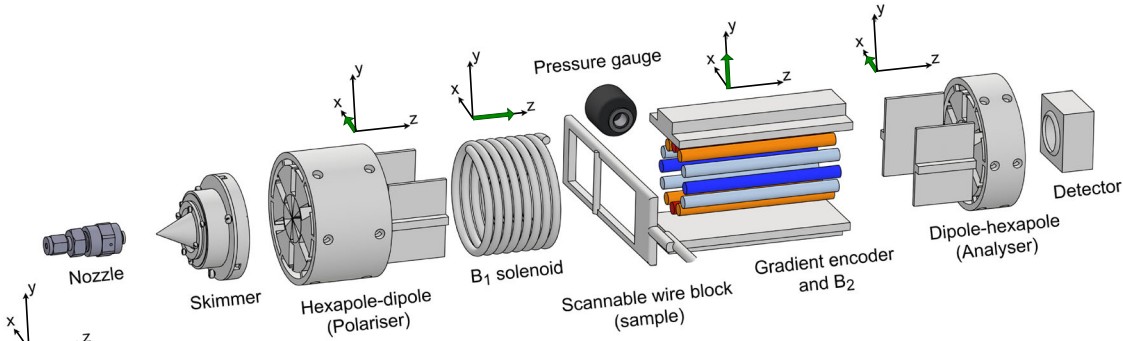

**Fig. 2 | Schematic of the experimental setup.** A cold nozzle followed by a skimmer is used to form the helium beam. A hexapole−dipole assembly is used as a spin polariser resulting in a beam polarised along the direction of the dipole field ($\hat{x}$). A homogeneous $\boldsymbol{B}_1$ field along $\hat{z}$ leads to Larmor precessions in the $xy$ plane used to both reverse velocity-dependent spin dephasing in $\boldsymbol{B}_2$ and to produce an additional 90° rotation for measuring both components of the complex signal. A moveable 100 μm wire was used to modify the beam profile imaged by magnetic encoding and also perform reference measurements of the beam profile. The encoding device uses a 12-wire geometry to produce a magnetic field along the $\hat{y}$ direction, which changes linearly as a function of the $x$ coordinate. The encoding device also includes a stronger homogeneous dipole field along the $\hat{y}$ direction, to eliminate the effect of the unwanted gradient field. A dipole−hexapole pair is used to focus particles with spins oriented along the $+\hat{x}$-axis towards a mass spectrometer particle detector at the end of the beamline. The field directions of the polariser (dipole), $\boldsymbol{B}_1$, $\boldsymbol{B}_2$ and the analyser (dipole) are illustrated above with green arrows.

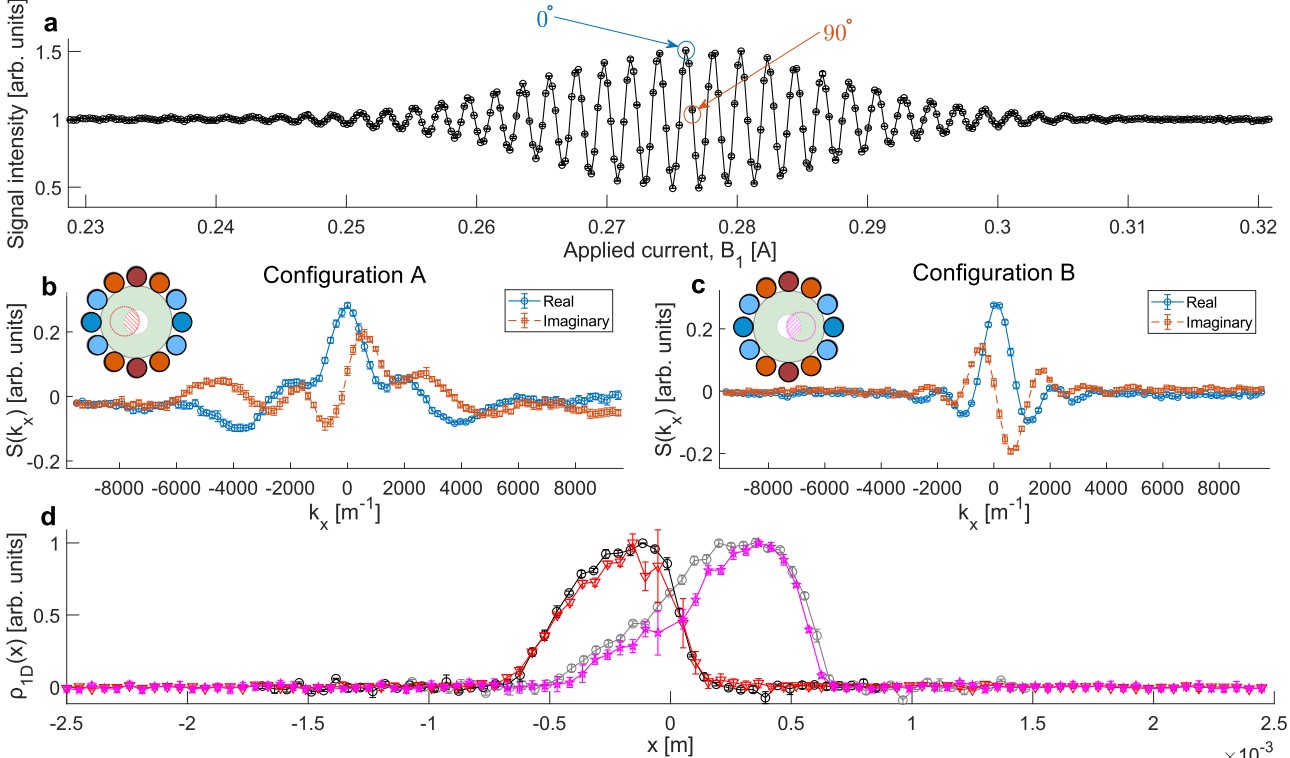

**Fig. 3 | Measured signals. a** Spin echo measurement performed by scanning the current (generating $B_1$) for a fixed $B_2$ value. The blue and orange arrows mark the currents used to measure the real (0°) and imaginary (90°) components of the signal. **b** and **c** $S(k_x)$ measurement for configurations A and B, respectively, after subtracting the constant signal level. The configurations are shown schematically in the inset plots. The red/magenta diagonal lines mark the overlap between the clear tube (white region) and the beam position in space (red/magenta circle), defining the shape of the beam going through. **d** $\rho_{1D}(x)$ reconstructions for configurations A and B (red triangles, magenta stars) along with reference wire-scan measurements for each configuration, respectively (black/grey circles). All error bars were calculated from the standard deviation of repeat measurements.

0° and 90° angles between the projection axes of the polariser and analyser

To image a beam with a non-uniform profile, we positioned the encoding device such that the hollow tube the beam can pass through partially overlaps the position of the beam in space (configuration A, illustrated schematically as an inset in Fig. 3b). The next step was measuring the signal while scanning $B_1$, this produces a spin echo signal shown in Fig. 3a, where the central maximum is the point where the magnitude of the field integral matches that of $B_2$ and cancels the coherency loss due to the velocity spread in the beam. The current producing the maximum and another displaced by $\frac{1}{4}$ of the oscillation period were identified as the 0° and 90° rotations used to measure the real and imaginary components of $S(k_x)$. Fixing $B_1$ at these values we measure the detector signal while scanning the current in the gradient wires of the encoding field. Figure 3b displays the real and imaginary components measured for configuration A. The red triangular markers in Fig. 3d present the magnitude of the Fourier transform of $S(k_x)$ and should correspond to the 1d spatial profile $\rho_{1D}(x)$ convoluted with the resolution function of the reconstruction. The encoder was then repositioned, allowing a different part of the beam through (configuration B shown schematically in the inset of Fig. 3c), producing $S(k_x)$ shown in Fig. 3c and the reconstructed profile shown by the magenta star markers in Fig. 3d. The block diagram shown on the left side of Supplementary Fig. S6 lists the steps we followed to obtain the profile.

Due to the discrete Fourier transform relation between $S(k_x)$ and $\rho_{1D}(x)$, the spatial resolution, $\Delta x$, and the total field of view (FOV) of the reconstructions presented in Fig. 3 are related to the $k_x$ values of the measurements through $\Delta x = (\max(k_x) - \min(k_x))^{-1}$ and FOV $= \Delta k_x^{-1}$ [26,27]. The $k_x$ values were evaluated using finite element calculations of the encoding device. For the currents used in the

experiment and an average beam velocity calculated from a Fourier transform of the spin echo measurement[13], the values for the measurements presented in Fig. 3b, c are $\Delta x = 52\mu m$ and FOV $= 5000\mu m$. To verify the validity of these calculations, we performed an independent reference measurement of the beam profile, by turning off the encoding field and scanning a vertical 100 μm-thick wire while measuring the increase in chamber pressure via the pressure gauge illustrated in Fig. 2. This wire scan measurement produces an inverse profile, i.e. positioning the wire into the beam increases the pressure in a way which is proportional to the number of particles which hit it at a particular $x$ position, and consequently had to be inverted to mimic the transmission profile $\rho_{1D}(x)$. The black and grey markers (for configurations A and B, respectively) in Fig. 3d show the inverted wire scans, which follow the magnetic encoding scans quite well for both configurations, confirming the validity of the reconstruction method and the scaling of $k_x$.

To further demonstrate the validity of the experimental resolution calculation, we also performed magnetic encoding measurements with the 100 μm-thick wire positioned within the beam (illustrated schematically in the inset plots in Fig. 4a and b). We repeated this for three wire positions, separated by 50 μm steps, for each of the encoder configurations. Figure 4a and b magnify the relevant part of the profiles reconstructed from an array of these measurements (plotted using the blue, orange and green markers) and show clear minima in all of these measurements, well resolved from each other and separated by the expected distance (50 μm). The magnetic encoding measurements that produced these profiles are shown in Supplementary Figs. S2 and S3.

Finally, to validate that the shape of the dip in the profile is what we expect to get for a 100 μm wire blocking the beam, we performed a

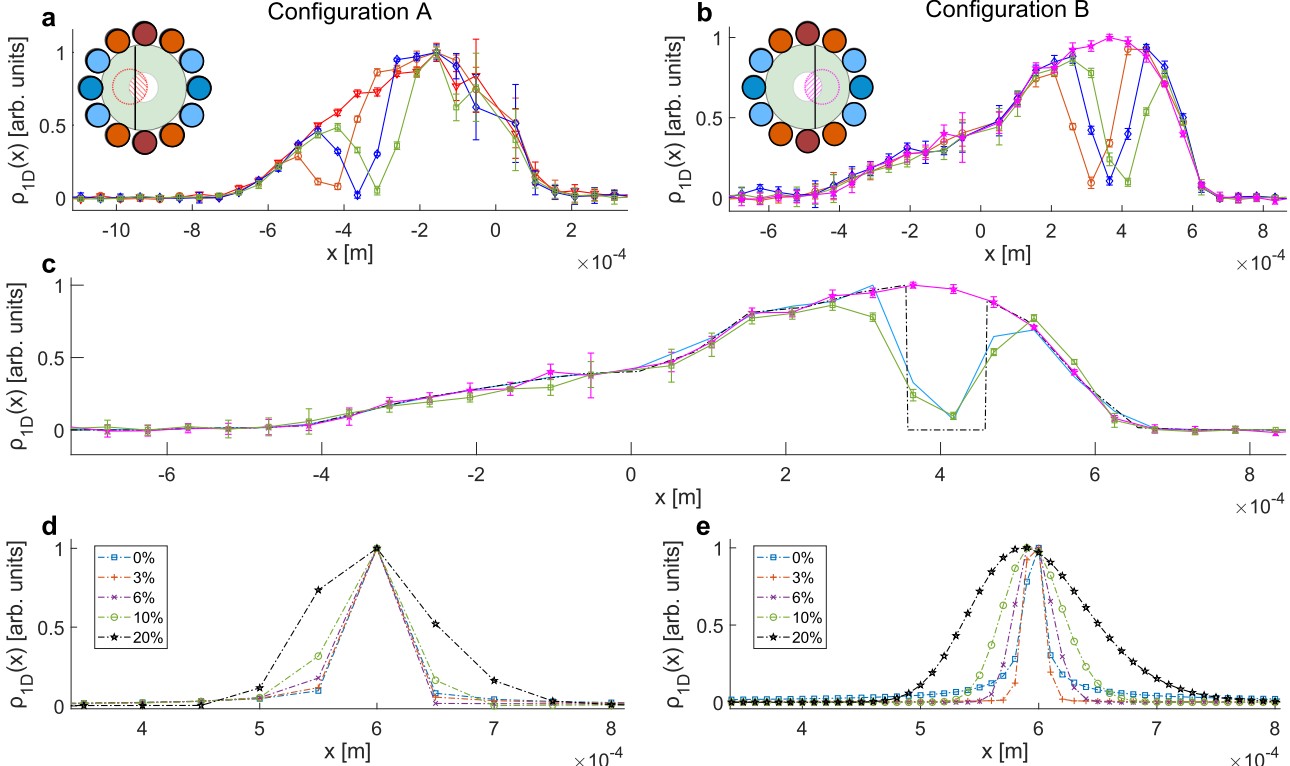

**Fig. 4 | Spatial profiles reconstructed from magnetic measurements. a, b** $\rho_{1D}(x)$ spatial reconstructions for configuration A/B without the wire and for three different wire positions. The signals were normalised to have the same magnitude at one position (−0.15 mm) where the wire does not obstruct the beam. The error bars were calculated from the standard deviation of repeat measurements. Note that the origin itself cannot be measured in a magnetic scan measurement due to uncertainty in the measurement background, which in a Fourier transform appears at the origin, furthermore the errors are larger in the points close to the origin due to slow drifts in the detection efficiency. **c** The initial beam density from a measurement without a wire (magenta markers) was multiplied by zero at the expected wire position to produce the input for the simulation (black dashed line). Precession calculations followed by a Fourier transform produced a simulated profile (blue markers) which compares very well with the profile reconstructed from a measurement (green markers). **d, e** Simulated reconstructions of $\rho_{1D}(x)$ for a sharp impulse peak centred at 0.6 mm and for resolutions of 50 and 10 μm, respectively, and for varying Gaussian velocity distributions with a FWHM of 0%, 3%, 6%, 10% and 20%.

simple numerical simulation of the experiment. For the initial input spatial distribution of the beam, $D(x)$, we used a spline fit of the measured magnetic profile without the wire in configuration A (magenta markers in Fig. 4c) and the effect of the wire was simulated via a transmission probability, $P(x)$, equal to zero within the 100 μm region occupied by the wire and unity elsewhere. The black dashed line in Fig. 4c shows the product of $D(x)$ and $P(x)$. We then calculated the spin precession of the beam through an ideal gradient field, Fourier transformed the result and obtained the simulated $\rho_{1D}(x)$ shown by the blue markers in Fig. 4c which is very similar to the profile reconstructed from the measurement (green markers) after applying the same scaling scheme.

**Scaling of acquisition time with the required resolution**
The scaling between the resolution of an image and the time it takes to acquire that image with an acceptable signal-to-noise ratio (SNR), differs significantly between the magnetic encoding imaging method and pin-hole microscopy. To make the comparison more useful, we will focus on 2d imaging.

In an optimally designed pin-hole microscope, reducing the size of the beam spot on the sample and correspondingly the pixel size by a factor of $N$, will reduce the flux by $N^4$ as shown by Bergin et al.[28] On the other hand, magnetic encoding experiments do not require multiple microscopic apertures or microscopic beam sources for their resolution, and increasing the resolution means the number of atoms contributing to a pixel will at the worst case (for a sample with a relatively flat/smooth structure, as will be discussed further below) reduce

linearly with each dimension, i.e. quadratically in 2d imaging. This is a direct consequence of the Fourier transform relations between the signal and the density function derived above and is explained and then demonstrated numerically below.

While in a SHeM experiment, the microscopic beam is scanned to measure each pixel separately, in magnetic encoding the signal of each $k_x$, $k_y$ measurement combines contributions from the entire macroscopic beam, which reaches the detector as can be seen from Eqs. (1) and (2) and illustrated in Supplementary Movie 1. The Fourier relation between the signal (reciprocal space) and the image (real space) we want to reconstruct, leads to three important properties: (1) The FOV of the image is inversely related to the interval between adjacent $k$ values, FOV $= \Delta k_x^{-1}$. (2) The pixel size of an image is inversely proportional to the range of $k$ values we scan, $\Delta x = (\max(k_x) − \min(k_x))^{-1}$, i.e. to increase the resolution of a magnetic encoding experiment, we need to continue to measure the signal for increasing $|k|$ values. (3) There is an inverse relation between the width of a feature in the image and the width of its corresponding signal. This is a well-known property of Fourier transform pairs, for example the Fourier transform of a gaussian with a width proportional to $\Gamma$ in real space will be a gaussian with a width proportional to $\Gamma^{-1}$ in reciprocal space[26]. Combining the last two properties means that the cost of enhancing the resolution of a magnetic encoding experiment in terms of measurement time and/or SNR depends on the shape of the sample we are imaging.

The inverse relation between the width in real and reciprocal space is demonstrated for two different types of 2d distributions in Fig. 5. Panel a shows a density distribution $\rho(x,y)_{smooth}$ containing a

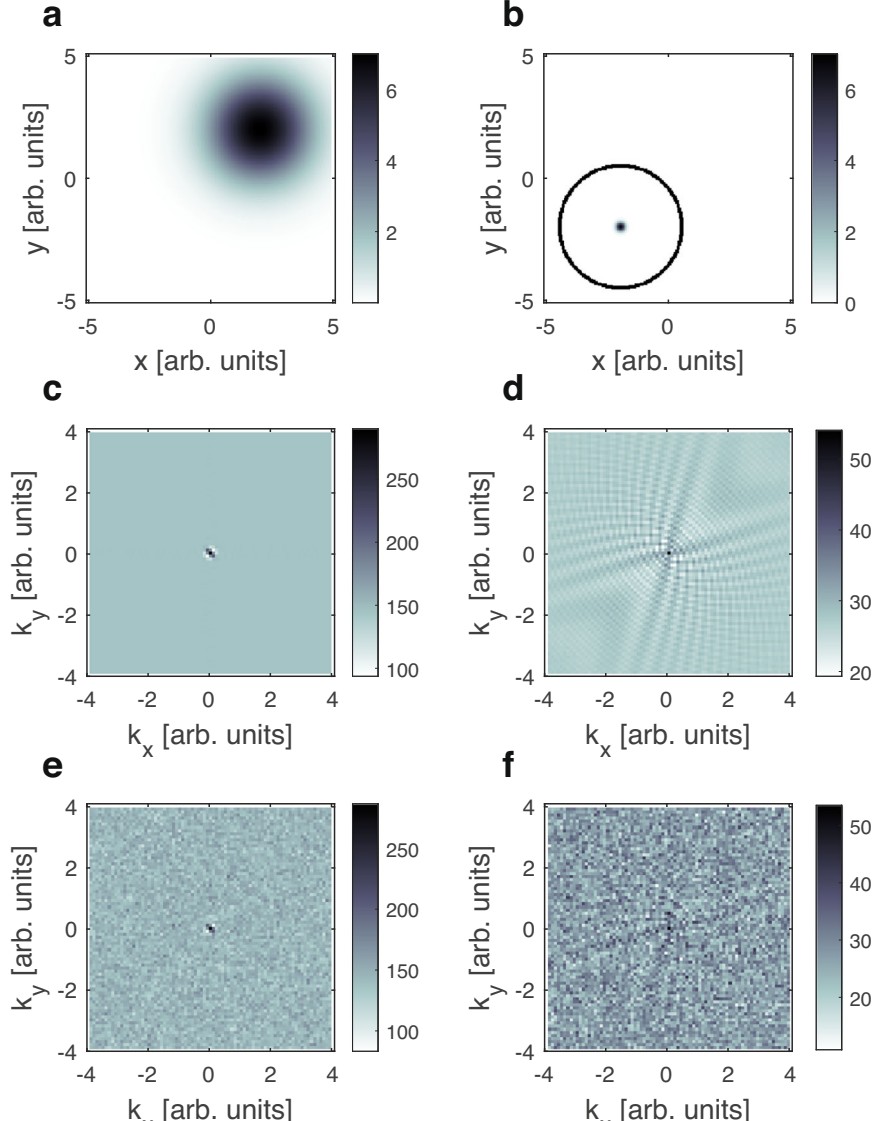

**Fig. 5 | Comparison of simulated 2d magnetic encoding signals for different types of density distributions. a, b** The original input spatial distributions $\rho(x,y)_{smooth}$ and $\rho(x,y)_{sharp}$ respectively. **c, d** The corresponding calculated signals $S(k_x,k_y)$ demonstrating how smooth/wide features in real space appear as signals concentrated near the origin of $k_x,k_y$, whereas sharp/narrow features in real space extend throughout the $k_x,k_y$ space. For clarity, just the real part of $S(k_x,k_y)$ is shown. **e, f** show the $S(k_x,k_y)$ plotted above them after adding random noise to mimic their appearance in the presence of shot noise in an experiment.

wide 2d Gaussian whereas panel b shows $\rho(x,y)_{sharp}$, a thin-walled high-intensity ring with a sharp peak at its center. Panels c and d show the 2d signals $S(k_x,k_y)$ obtained when calculating Eq. 2 using $\rho(x,y)_{smooth}$ and $\rho(x,y)_{sharp}$. For $\rho(x,y)_{smooth}$ the signal is confined to a very narrow region around the centre, beyond which we get a constant background level, whereas for $\rho(x,y)_{sharp}$ the signal is characterised by small-amplitude oscillations which persist all the way to the edge of the 2d $k$ space.

In order to assess the impact of noise on imaging the two $\rho(x,y)$ distributions shown in panels a and b, we added random noise to their calculated signals. The noisy signals shown in Fig. 5e and f were generated by adding to each $S(k_x,k_y)$ value a random number from a Gaussian distribution with a standard deviation equal to $\sqrt{S(k_x,k_y)}$. This type of noise mimics the fluctuations expected for a signal dominated by shot noise.

Figure 6 shows how increasing the spatial resolution by a factor of $N$ affects the SNR of a reconstructed image. The upper row of panels shows the mesh of $k_x,k_y$ points needed for different resolution enhancement factors of 1, 2, 4, 6 and 8, starting with a grid of 10×10 for the lowest resolution. We chose to compare these different resolutions while keeping the FOV constant which means a fixed interval between neighbouring $k_x,k_y$ points. We have also chosen to simulate a scenario where the total time for measuring the five different resolutions is kept the same. To keep both the FOV and the overall measurement time fixed while increasing $N$ means that the time devoted to measuring each specific $k_x,k_y$ value needs to be reduced by a factor of $N^2$. To mimic this reduction of time for the individual measured values, we divided each cell of $S(k_x,k_y)$ by $N^2$ before adding the corresponding shot noise.

The middle and bottom rows in Fig. 6 show the reconstructions obtained from inverse Fourier transforms of the simulated signals for resolution enhancement factors of 1, 2, 4, 6 and 8, where the panels in the middle and bottom row correspond to $\rho(x,y)_{smooth}$ and $\rho(x,y)_{sharp}$, respectively. The difference between the two cases is quite striking. Increasing the resolution (while keeping the FOV and overall

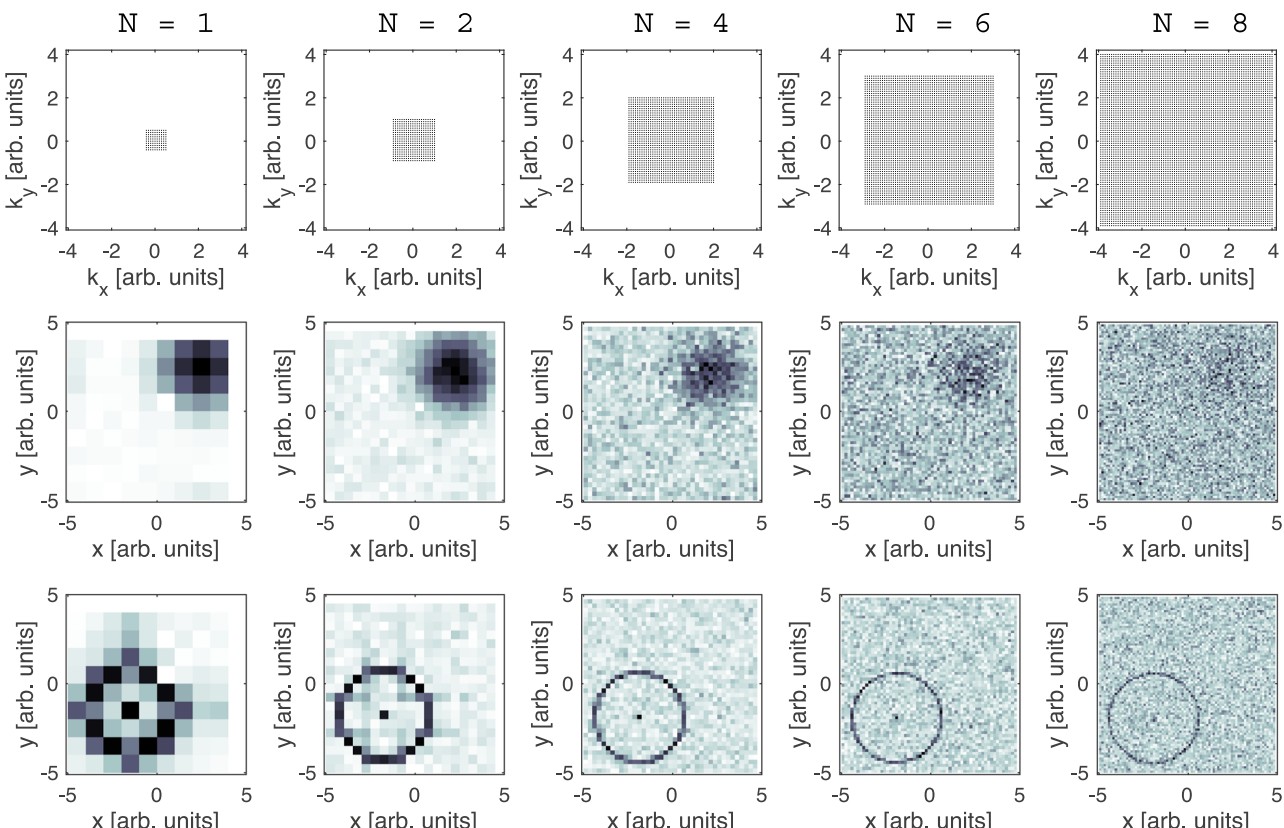

**Fig. 6 | Simulating the effect of resolution enhancement on the SNR of images reconstructed using magnetic encoding.** The top row (reciprocal space) illustrates the $k_x, k_y$ measurement points required for 5 different resolution enhancement factors ($N = 1, 2, 4, 6, 8$) corresponding to meshes with $10 \times 10$ (left side) up to $80 \times 80$ (right side) elements. The middle and lower rows (real space) show reconstructions of $\rho(x,y)_{smooth}$ and $\rho(x,y)_{sharp}$, respectively, for the five different resolution enhancement factors. The colours were scaled between the minimum and maximum values for all images.

measurement time fixed) leads to a clear reduction in SNR for the case of the wide Gaussian distribution (middle row in panel 6), with the feature essentially disappearing at the highest resolution. In contrast to the distribution with the sharp features, initially, the SNR seems to improve and only deteriorates at the highest two resolutions, but even then, the feature is still quite clear. This difference in how the SNR deteriorates with resolution can be understood if we consider the effect of adding more $k_x, k_y$ measurements. Since $S(k_x, k_y)$ is concentrated at the centre for the case of $\rho(x,y)_{smooth}$ (Fig. 5c) measuring further out in $k_x, k_y$ essentially just adds noise. In contrast for $\rho(x,y)_{sharp}$, (Fig. 5d) measuring further includes both noise and signal, and the Fourier transformation used to reconstruct the final image has an averaging effect. If the resolution is further enhanced such that the pixel size is comparable to the feature width, the signal decays and the SNR reduces. For completeness, the simulations in Fig. S5 show the case where the individual measurement time of each $k_x, k_y$ measurement is fixed rather than the overall measurement time, leading to a much-improved SNR, nevertheless the differences between smooth and sharp $\rho(x,y)$ are still clearly apparent.

## Discussion

The proof of principle measurements and the numerical simulations presented above demonstrate the validity of the magnetic encoding approach. To create high-resolution microscopes that are based on magnetic imaging, various substantial experimental developments will be needed. Below we discuss the next development steps as well as the longer-term prospects of the technique in terms of its expected contrast and resolution capabilities.

For most applications, 2d imaging is essential. While conceptually, this is a rather straightforward extension of 1d measurements, it

will require building a dedicated instrument with space for the additional imaging elements. One way to image the beam in 2d is to add a second perpendicular magnetic gradient device, allowing us to independently scan both $k_x$ and $k_y$ in Eq. (2), this is the scheme that was simulated in Figs. 5 and 6 and is described as a block diagram in Supplementary Fig. S6. Another approach would be to use just one gradient field and rotate its orientation with respect to the sample, similar to what is done in projection reconstruction MRI[11].

While the lateral resolution of existing SHeM setups is on the order of micrometres, the very short deBroglie wavelength of the particles and the fact they interact with the outermost electrons of the surface, means that SHeM has been successfully used to identify nano-metre topographic features which are orders of magnitude smaller than its lateral resolution, including differentiating the shape and chemical identity of single layers of adsorbed particles and even identifying local atomic-scale crystalline order seen in diffraction imaging mode[2,29,30]. Since the interaction of $^3$He and $^4$He atoms with surfaces is very similar, we expect all the contrast mechanisms and topographic resolution observed in SHeM to apply to magnetic encoding imaging. A new contrast mechanism might become available with magnetic encoding imaging if the surface itself is magnetic and alters the spin state of the beam particles during scattering, thus relating the image of the surface to local magnetic properties. To the best of our knowledge, nuclear spin flips have never been observed in $^3$He scattering from non-magnetic surfaces, and until measurements showing spin−flips from magnetic surfaces become available, this potential contrast remains hypothetical.

To assess the expected spatial resolution of magnetic encoding microscopy, it is useful to rewrite the resolution criteria written above

explicitly, $\Delta x \propto \frac{|\bar{\mathbf{v}}|}{\gamma L \max(\frac{dB_y}{dx})}$, and compare the different parameters with those used in our $\Delta x = 50\,\mu m$ proof of principle measurements. Reducing the velocity by about $\sqrt{2}$ is what we can expect from cooling the nozzle from 40 to 20 K (supersonic expansion of $^3$He is a challenge below 20 K), further slowing down could be achieved using a heavier buffer gas in the beam (see for example, the slowing of water beams with krypton[31]), although this would be on the expense of the $^3$He flux. The length of the encoding device used in this study was restricted by constraints of the MMI apparatus and could be made longer. A gain of approximately half an order of magnitude seems reasonable before the instrument becomes unpractically long. The maximal gradient of the encoding device is where the biggest improvement can be achieved, as we used a very modest gradient field in the proof of principle experiments. Relatively straightforward improvements would be adding cooling to allow higher currents and reducing the dimensions of the device, both of which will increase the gradient and together can be expected to improve the resolution by about one order of magnitude. Further enhancement of the gradient field could be possible using gradient field designs which are based on superconductor coils or by incorporating permanent magnets. An example of the latter is the hexapole polariser of the MMI setup, which has magnetic field gradients that are three orders of magnitude larger than that of the current encoder field. However, adjusting the field strength to scan $k$ when using a device that includes permanent magnets would be a technical challenge. Finally, while the gyromagnetic ratio $\gamma$ is a fixed property of $^3$He, using beams of excited atoms (e.g. metastable helium) or paramagnetic molecules ($O_2$) would increase the gyromagnetic ratio by about 3 orders of magnitude. This change would also open up new contrast mechanisms due to the different interactions with the surface, but would, on the other hand, be a compromise in terms of the inert and gentle properties of the particle beam probe.

Implementing a combination of some of the more straightforward changes should quite readily allow resolutions on the order of a micrometre. If we also allow for much stronger gradient devices and/or use a different particle, such as metastable helium or oxygen, a simple multiplication of the factors mentioned could lead to optimistic predictions of nano-metre lateral resolution, however, there are several issues that will make this very difficult if not impossible and need to be researched further before such a claim can be made.

One limitation is that the spread in velocities of the particle beam will lead to a deterioration of the resolution and blurring of sharp images. The use of an average velocity in the definitions of $k_x$ and $k_y$ does not take into account the different velocities within the beam. Reconstructed profiles shown in Fig. 4d and e illustrate this limitation. Numerical simulations were used to calculate the signals and reconstructed spatial profiles of a sharp impulse peak, for velocity gaussian distributions of different widths. When imaging with 50 μm resolution (Fig. 4d) the velocity broadening effect is only noticeable for very wide velocity distributions, whereas for 10 μm (Fig. 4e) the broadening effects become noticeable already for beams with an FWHM of a few percent. One solution is to use better supersonic expansion conditions, which can produce velocity distributions with sub-percent FWHM[18] and should be sufficient for resolutions on the order of a micro-metre. To further improve the resolution beyond this would probably require resolving the different velocities in the beam, this should be possible either by using pulsed / chopped sources combined with time of flight detection[32], or by using spin echo techniques[13], which will need to be developed and incorporated in the imaging scheme.

A second limitation is that the same changes mentioned above for improving the resolution, also increase the deflection force on the particles passing through the encoder. For resolutions that are smaller than a micrometre, the deflection becomes significant and can deteriorate the image. Compensation schemes/geometries that cancel or reduce the overall deflection will need to be developed to achieve resolutions of hundreds of nanometres or better.

The third limitation will be the signal-to-noise ratio and the time that can be realistically devoted to the imaging process. On the one hand, the much weaker scaling of the SNR with the resolution that magnetic encoding has, in comparison to pin-hole microscopy, should make it easier to keep improving the resolution. On the other hand, the beam flux at the sample, the detection efficiency, the reflectivity of the surface (in reflection mode) and how strong the contrast is, could all potentially become limiting factors. Before a dedicated magnetic encoding microscope is designed, built and tested on various samples, it is impossible to quantify all of these factors. We note, however, that in our proof of principle experiments, we used a commercial mass spectrometer which is ~5 orders less sensitive than state-of-the-art large ionisation detectors[33]. We also used a large (2 mm) beam instead of a brighter smaller beam source, which would be better suited for a microscopy experiment[28]. Consequently, there is room for huge improvements by implementing already established technological improvements, making magnetic imaging a promising method for imaging with neutral beams with resolutions comparable to and potentially even better than those available with current SHeM. The advantage of magnetic encoding over pin-hole technology in terms of the measurement time, is expected to be especially pronounced when imaging sharp features within a wide FOV, as demonstrated in Fig. 6. We would also like to stress that while the magnetic encoding method could be applied on its own to achieve spatial resolution, it can also be used as an add on to pin-hole or focusing microscopes, providing an additional boost to the resolution with a much smaller price in terms of measurement time and with a resolution change which simply requires enhancing the gradient scan range rather than changing the imaging aperture.

Finally, the magnetic encoding technique is not restricted to Helium-3. In theory, any atom or molecule with an electronic, nuclear, or rotational magnetic moment can be manipulated to resolve the spatial position using the schemes presented above. However, coupling between the different magnetic moments and multiple overlapping oscillation frequencies can make the analysis far more complicated[15,24]. A relatively simple case is using $D_2$ molecules, where it has been shown that the non-rotating quintet $I = 2, J = 0$ state, which dominates a cold $D_2$ beam, can be magnetically manipulated in a way that is analogous to experiments with $^3$He[34]. In addition to the fact that $D_2$ is a more readily available and less expensive gas than $^3$He, it also scatters from surfaces differently[35], leading to different image contrast and complimentary information to an image produced using a helium beam.

## Methods

Performing magnetic phase encoding requires prior characterisation of the fields generated by the gradient coils in order to accurately scale both the axes of Fourier space and by extension, the spatial coordinates that are reconstructed. The encoder geometry was designed to fit the physical constraints of the existing MMI setup, have an acceptable degree of gradient inhomogeneity within the region the beam passes through and allow us to perform proof of principle 50 μm resolution profile measurements.

As we are using electromagnets for the encoding device it is useful to define $k$ per unit current, $k_x^I(x,y)$, related to the field per unit current $B_y^I$ through

$$k_x^I(x,y) = \frac{\gamma}{2\pi|\bar{\mathbf{v}}|} \int \frac{dB_y^I(x,y,z)}{dx} dz. \qquad (3)$$

where we integrate along the beam propagation direction instead of multiplying by the length of the encoding device to account for field changes at the edges of the device. Multiplying $k_x^I(x,y)$ by the current

$I_0$ passed through the wires and a specific position $x$ gives us the total spin phase accumulated over the traversal of the encoding device.

A finite-element simulation software (ANSYS Maxwell) was employed to evaluate $B_y^I(x,y,z)$ of the encoding device, composed of 12 (150 mm long, 1.05 mm diameter) wires spherically distributed around a diameter of 6 mm and with current polarities detailed pictorially in Supplementary Fig. S4. The calculated values for the average gradient and, therefore, average $k_x^I$ within a diameter of 3 mm centred around the beam axis were 0.134 Tm A$^{-1}$ and 857 m$^{-1}$A$^{-1}$ respectively. Using the average $k_x^I$ we calculated that a maximum current of 11.2 A and a current spacing of 0.233 A will lead us to a pixel resolution of ~52 μm and a FOV of 5000 μm. The determination of $k_x^I(x,y)$ also allows for a quantitative evaluation of the degree of gradient field inhomogeneity as a function of both spatial positions perpendicular to the beam propagation axis (i.e. $\hat{x}$ and $\hat{y}$), and additionally includes the effects of the non-symmetric edge connections for each of the 12 wires. The results are shown in Supplementary Fig. S1.

The field integral of $\mathbf{B_2}$, was also modelled and optimised through minimisation of $B_y$ inhomogeneity across the beam-occupied region using the same finite element software, the results of which were then applied to estimating the spin echo condition with the $\mathbf{B_1}$ solenoid field, leading to the identification of the region measured in Fig. 3A and therefore the currents corresponding to the aforementioned real (0°) and imaginary (90°) rotations. The desired strength of the axis selecting field (of approximately 170 G, for a 90% retention of polarisation in the measurement plane) was achieved with the use of a high permeability soft magnetic iron (ASTM A848) to ensure a strong flow of magnetic flux via the yoke (illustrated by the green arrows in Supplementary Fig. 4) transmitted over the 12 mm gap separating both poles.

The simulated profiles in Fig. 4c–e were calculated using MATLAB code to solve the evolution of classical spin-phase when subject to ideal gradient fields (i.e. $\mathbf{B}_{\mathrm{encoding}} = (0, \frac{dB_y}{dx}x, 0)$) and include a summation over both the distribution of spatial positions and velocities of the simulated beam.

### Details of the molecular beam

The $^3$He beam was formed through the supersonic expansion of 1 bar through a 30 μm diameter nozzle cooled to 40 K. As $^3$He is a very expensive gas, a recycling system is used to pump, clean and recompress the gas back and redeliver it to the nozzle. By Fourier transforming a $B_1$ scan measurement, we determined the velocity distribution of the $^3$He beam[13], which can be approximated as a Gaussian peak with a central velocity component of $|\bar{\mathbf{v}}| = 756.6$ m s$^{-1}$ and full-width half maximum of 52.2 m s$^{-1}$.

## Data availability

The measured and simulated data generated in this study have been deposited in the GitHub repository under accession code https://github.com/Mog-Lowe/Magnetic-beam-spin-encoding-NCOMMS-24-03503-. (https://doi.org/10.5281/zenodo.12707070).

## Code availability

The Matlab scripts used in the manuscript and supplementary information are available at https://github.com/Mog-Lowe/Magnetic-beam-spin-encoding-NCOMMS-24-03503-. (https://doi.org/10.5281/zenodo.12707070).

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

## Acknowledgements

This work was funded by an ERC consolidator grant, Horizon 2020 Research and Innovation Programme grant 772228 (G.A.) and an EPSRC standard grant (EP/X037886/1) (G.A. and H.C.). G.A. is grateful to Prof. Aharon Blank and Prof. Marcel Utz for stimulating discussions. The authors are grateful to Mr. Paul Smith, Mr. Philip Hopkins and Mr. Kevin Morgan for their valuable technical support.

## Author contributions

G.A. conceived and supervised the project. M.L. designed, assembled and tested the magnetic field encoder and wrote the analysis code. Y.A. designed, installed and characterised the 3rd arm of the MMI apparatus. M.L., H.C. and G.A. performed the measurements, interpreted the results and prepared the manuscript. All the authors read and commented on the manuscript.

## Competing interests

G.A. is the inventor of a patent application submitted by Swansea University that covers the use of magnetic encoding for neutral beam microscopy. The remaining authors declare no competing interests.
