## [Peer Review File · Nature Communications]

Neutral beam microscopy with a reciprocal space approach using magnetic beam spin encodingREVIEWER COMMENTS

Reviewer #1 (Remarks to the Author):

Journal: Nat Comms

Title: Magnetic beam spin encoding; a reciprocal space approach for Neutral beam microscopy.

Authors: Morgan Lowe, Yosef Alkoby, Helen Chadwick, and Gil Alexandrowicz

This paper reports a new approach to SHeM imaging through reconstruction of ^3He scattering distributions through manipulation of magnetic spin moments.

The idea and experimental implementation of this new approach is very interesting and warrants publication in Nature Communications. This paper represents a novel implementation of the use of magnetic spins to deliver contrast in scanning helium microscopy (SHeM). The concepts outlined in the paper are complex and the manuscript needs some work to clarify the arguments that the authors are trying to convey.

Specifically, the following points should be addressed.

1. General: The pages of the manuscript should be numbered! It makes it very difficult to reference sections for comments without page numbers. I will define the first page with the abstract as page 1 for the purposes of the rest of this report.
2. P2 L10: The authors should note that the first 2D images (ref 7) were “albeit for a transmission mask”.
3. P2 L34: The authors state that ^3He is “particularly relevant for SHeM applications”. However, the other SHeM instruments that are referred to are all based on ^4He . The authors need to discuss the implications for the application of ^3He to the development of SHeM, particularly with regards to: (1) cost, (2) availability, and (3) practicality.
4. P3 Fig 1: This figure is not very clear and could provide a better illustration of the encoding scheme than it currently does. In particular:
 - a. The red arrows are very unclear, especially when printed – need to make this part of the figure clear.
 - b. It would be helpful to have the B-field included in the figure – could then be referenced in the text.
 - c. The relationship between the magnetic moment diagrams before and after the sample interaction is not clear – what is the diagram trying to tell me?
 - d. Include the spin analyser and detector in the figure – then use this to illustrate the complex signal discussion in the text.
 - e. What is $\rho(x,y)$ in the figure? Is it the cyan rectangle?
5. P3 Eq 1: The authors should provide more detail for the derivation of equation 1 – could be placed in supplementary information.
6. P4 8: The authors state that a 2D image can be obtained by “simply adding a second encoding field along the beam line which produces a gradient along the y coordinate.” This doesn’t sound that simple! What would be the challenges of associated with achieving 2D imaging this way?
7. P5 Fig 2: Again, this is another figure that needs more work to better illustrate the complex concepts that are being described. In particular:
 - a. Put axes on both the field distribution patterns and the magnet assembly figure. Relative to Fig 1 which way are the axes now?
 - b. Show the ^3He beam direction in the figure – presumably it passes through the ceramic tube.
 - c. Why are there multiple ceramic tubes? Why is the inner one only 1.5 mm in diameter?
 - d. Show figures for the field directions with and without B_2 ; what is the relative size of B_2 ?
8. The authors state that the expression that is given for k_x , and k_y , “are not accurate” and that there is a velocity dependence of the phase. But there is already a $1/v$ dependence of the k_i on velocity, so what is the correct expression for the wavevectors? The authors need to clarify this point.

9. P6. The discussion around the refocusing field is especially unclear and could do with more explanation. It would be helpful to add a supplementary section that includes a picture of the beamline containing both B_2 and B_1 and a description of the refocusing process. For example, how is it that the signal intensity is a measurement of the “magnetic field integral”? How do we know that the peak in the oscillation amplitude is the 0° phase point (and likewise that the first zero crossing is the 90° phase point)? I think that it would be good to help the reader out here with a concise explanation of the process that doesn't require them to trawl through multiple references!
10. P7: If I understand this next section correctly, the authors now describe two configurations whereby the beam is blocked by a “sample” located in two positions. The real and imaginary magnetic field integral data ($S(k_x)$) is then collected at two magnetic field values of B_1 corresponding to the 0° and 90° phase conditions. This data is then Fourier transformed into the density distribution function. In the case of no sample, this is the beam profile. Again, this section is really challenging to understand and should be restructured for clarity:
 - a. The authors should provide a picture showing the A and B configurations.
 - b. P7 L8: Why is the encoding device only positioned through part of the beam? Why does moving the encoding device lead to an apparent shift of the beam profile? What is the actual beam profile – is it the sum of the two profiles shown in Fig 3a? Are the profiles shown in Fig 3a the convolution of the actual beam profile with the response function of the encoding device?
 - c. It would be helpful to have a figure that shows the workflow in getting the final image from the data (i.e. Figure 3b to Fig 4c and Fig 3a to Fig 4b).
 - d. It would be helpful to label the two configurations in Fig 3a.
 - e. Just a minor point, why label config A on the right and config B on the left in fig 4, and the reverse in Fig 3b and 3c? Its just confusing and unhelpful. It would make more sense for the labels to read A to B left to right in both cases.
 - f. In my view, it would be better to have Fig 3b and Fig 3c in the same figure as profiles shown in Fig 4a – then the reader can see how one flows into the other.
11. P8: There is no discussion of Fig 5b and Fig 5c in the text?
12. P12-13: The discussion regarding the potential imaging time saving with increasing resolution is interesting. However, it is not clear in this comparison as to whether there is a difference in the signal to noise between the pinhole approach and the magnetically encoded approach. For example, what are the relative signal strengths between the two approaches? How would we actually compare between the two?

In addition, the following minor revisions should be addressed:

1. P6 L5: Provide a reference for the statement: “typically a FWHM of a few percent in supersonic beams”
2. Could the authors have a consistent x-axis distance scale for all of the figures.
3. Could the authors make all of the axis labels and titles much larger!
4. P13 L20/21: Correct “loses” to “losses”.

Reviewer #2 (Remarks to the Author):

Lowe et al. present the fascinating idea of performing MRI-style imaging experiments of atom-surface scattering to provide a new modality for surface microscopy. The work builds on ^3He spin-echo experiments that are now well established and provide unique insights into surface dynamics. The methodology is sound and the experiments have the potential to be of interest to the readership of Nature Comms. That said, I think that the work described is suited to a more specialist journal.

Advances in microscopy normally relate to improvements in resolution or contrast. Even new techniques need to demonstrate or promise an advance in one of these aspects. Here, the spatial resolution is 50 microns, which is not competitive for neutral atom microscopy. Unfortunately, there is no clear projection for what resolution might be obtained with the new technique. (Is one limiting factor, as in MRI, the maximum B fields and gradients that can be achieved?) Although contrast is mentioned in the introduction, it is not described in detail, so it is unclear what the new technique might teach us about the atom-surface interaction or what new samples could be imaged. The promise of this new technique is thus unclear.

I am not sure that the manuscript will be accessible to readers outside the (small) atom scattering community. This isn't essential for publication in Nature Comms but I think that it would be sad for the wider microscopy community not to appreciate the significance of the work. For example, the apparatus description needs a schematic. The discussion surrounding figure 6 is hard to follow. The description of the signal levels scaling with the '4th power of the pin hole dimension' will be unfamiliar to microscopists who are more used to descriptions of beam brightness and optically-limiting apertures. There is a comment in the final sections relating to the detector efficiency that will be unclear to most readers. How atoms scatter and depolarise during scattering, and if they still contribute to the detected signal isn't obvious. There are many such minor issues throughout the manuscript.

Overall, I think that this is lovely work and a substantial technical achievement but not yet suitable for this journal. The promise of the new technique needs to be explained in greater detail and the description needs to be more accessible.

Reviewer #1 (Remarks to the Author):

This paper reports a new approach to SHeM imaging through reconstruction of ^3He scattering distributions through manipulation of magnetic spin moments.

The idea and experimental implementation of this new approach is very interesting and warrants publication in Nature Communications. This paper represents a novel implementation of the use of magnetic spins to deliver contrast in scanning helium microscopy (SHeM). The concepts outlined in the paper are complex and the manuscript needs some work to clarify the arguments that the authors are trying to convey.

We would like to thank the reviewer for their detailed and very useful report. We accept the reviewers view that there was need to clarify some of the arguments presented in the paper. We have changed the manuscript quite substantially, adding explanations in both the main text and the supplementary information, additional figures, animations and block diagrams all of which are described in more detail below. Red text is used below to mark the reviewers comments, whereas our answers and specific changes to the manuscript appear as black and blue text correspondingly .

Specifically, the following points should be addressed.

1. General: The pages of the manuscript should be numbered! It makes it very difficult to reference sections for comments without page numbers. I will define the first page with the abstract as page 1 for the purposes of the rest of this report.

Changes: Page numbers added.

2. P2 L10: The authors should note that the first 2D images (ref 7) were “albeit for a transmission mask”.

Changes: A note that these were for a transmission mask was added.

3. P2 L34: The authors state that ^3He is “particularly relevant for SHeM applications”. However, the other SHeM instruments that are referred to are all based on ^4He . The authors need to discuss the implications for the application of ^3He to the development of SHeM, particularly with regards to: (1) cost, (2) availability, and (3) practicality.

The sentence the reviewer refers to seems to have been misleading. The original sentence was meant to compare helium-3 to other atoms or molecules, so when we wrote particularly relevant it was in the sense that helium-3 is similar to helium-4 in terms of its inertness and surface sensitivity. We have clarified this sentence. Regarding the cost and availability of helium 3, the gas can be recycled very efficiently (as it is in our system). We have added a note to this in the methods section. We have also emphasised that helium-3 is expensive and not readily available in the last paragraph of the paper when comparing with D_2 .

Changes: The paragraph on page 2 was changed and now reads “As a first demonstration of the methodology, we chose a beam of ^3He atoms. Low energy ^3He beams are similar to the ^4He beams used in SHeM applications in terms of surface sensitivity and inertness, while having a non-zero magnetic moment which is essential for magnetic encoding imaging.”

The last paragraph of the paper was changed and now reads “In addition to the fact that D_2 is a more readily available and less expensive gas than ^3He , it also scatters from surfaces differently³⁵, leading to different image contrast and could supply complimentary information to an image produced using a helium beam.”

Changes: A sentence was added in the methods section “The ^3He beam was formed through the supersonic expansion of 1 bar through a $30\mu\text{m}$ diameter nozzle cooled to 40K . As ^3He is a very expensive gas, a recycling system is used to pump, clean and recompress the gas back and redeliver it to the nozzle.”

4. P3 Fig 1: This figure is not very clear and could provide a better illustration of the encoding scheme than it currently does.

We accepted this criticism, and have substantially altered the figure. It is now a sequence of snap shots describing the evolution of the magnetic moments. We have also included an animation in the supplementary information section SV1 which should make it easier to understand the principle of the technique. We have also located the interaction with the sample before encoding. While for explaining the method it can be positioned before or after the encoder, we decided it will be easier if it was consistent with the actual experimental system which now has a separate schematic (new figure 2).

In particular:

a. The red arrows are very unclear, especially when printed – need to make this part of the figure clear.

Changes: The visualisation of magnetic moments was changed to make sure they aren't interpreted as a representation of the magnetic field. We added animation SV1 to help show the dynamics in a better, more understandable way.

b. It would be helpful to have the B-field included in the figure – could then be referenced in the text.

We tried this but found the result confusing. Hopefully the axes drawn in the new figure 1 and the accompanying text makes its direction clear.

c. The relationship between the magnetic moment diagrams before and after the sample interaction is not clear – what is the diagram trying to tell me?

Changes: See answer to comment 4a.

d. Include the spin analyser and detector in the figure – then use this to illustrate the complex signal discussion in the text.

Again, we tried this suggestion, but found the result potentially more confusing (too many concepts in one figure). We have instead added more detailed explanations of the real and imaginary parts of the signal in the supplementary information section (S7), which hopefully clarify this issue.

Changes: Detailed explanations added in supplementary information (S7).

e. What is $\rho(x,y)$ in the figure? Is it the cyan rectangle?

Changes: New figure 1 contains a blocking object (sample) which is clearly indicated by a label and arrow.

We also added a note in the caption to say that the blocking object defines $P(x,y)$. "The beam interacts with a sample which blocks some of the trajectories, resulting in $P(x,y)$ being zero for specific x,y positions"

5. P3 Eq 1: The authors should provide more detail for the derivation of equation 1 – could be placed in supplementary information.

Changes: A more detailed derivation of equations 1 and 2 has now been added to the supplementary information (S7).

6. P4 8: The authors state that a 2D image can be obtained by "simply adding a second encoding

field along the beam line which produces a gradient along the y coordinate.” This doesn’t sound that simple! What would be the challenges of associated with achieving 2D imaging this way?

We accept the reviewers comment, we have reworded the sentence and taken out the word simply. We have added a more detailed description in the supplementary information

Changes: New sentence on page 4 “Such an image can be obtained by adding a second encoding field along the beam line which produces a gradient along the y coordinate”.

Changes: Explanations added in the supplementary information (S7).

7. P5 Fig 2: Again, this is another figure that needs more work to better illustrate the complex concepts that are being described.

In response to comments 7 and 9 we have added a new figure 2 to the main manuscript to illustrate the measurement setup and moved the detailed description of the encoding magnet assembly to the supplementary information section (S4) where we can discuss the details the reviewer suggests without overloading readers which might not want to go into this level of detail.

In particular:

a. Put axes on both the field distribution patterns and the magnet assembly figure. Relative to Fig 1 which way are the axes now?

Changes: Axes have been added.

b. Show the ^3He beam direction in the figure – presumably it passes through the ceramic tube.

Changes: direction of beam with respect to the encoder can now be seen in the new figure 2.

c. Why are there multiple ceramic tubes? Why is the inner one only 1.5 mm in diameter?

The outer tube supports the wires which produce the magnetic field. The internal tube was inserted to limit the beam to the central 1.5mm diameter region where the homogeneity of the magnetic field is good enough to not have any noticeable effect on the accuracy of the reconstruction. Combining multiple tubes were just our way of getting the desired external and internal diameter, we removed this from the new figure to avoid the confusion it causes.

Changes in the text: Figure caption (S4) was changed “The inner diameter of the ceramic tube (1.5mm) was chosen to limit the beam passing through the encoding device to a region where the gradient is sufficiently homogeneous.”

Changes in figure S4: Multiple ceramic tubes made into one.

d. Show figures for the field directions with and without B2; what is the relative size of B2?

Changes: Figures of the field directions with and without B2 have been added to figure S4.

8. The authors state that the expression that is given for k_x , and k_y , “are not accurate” and that there is a velocity dependence of the phase. But there is already a $1/v$ dependence of the k_i on velocity, so what is the correct expression for the wavevectors? The authors need to clarify this point.

The equations for k_x and k_y are written with the average velocity, this means that individual particles with slightly larger or smaller velocities will accumulate slightly different phases. The sentences describing this have been reworded and now appear in page 13 when discussing the results shown in figures 4d and 4e.

Changes in the text: Reworded sentence which now appears in page 13 "...The use of an average velocity in the definitions of k_x and k_y does not take into account the different velocities within the beam. Reconstructed profiles shown in figures 4d and 4e illustrate this limitation."

9. P6. The discussion around the refocusing field is especially unclear and could do with more explanation. It would be helpful to add a supplementary section that includes a picture of the beamline containing both B2 and B1 and a description of the refocusing process.

We have added a schematic drawing (fig 2 in the main text) which should make it easier to understand how the measurements were performed. We have also added an animation SV2 which demonstrates visually how the velocity spread can be refocused (partially) even when the fields are perpendicular.

Changes: Added new figure 2 and supplementary animation SV2.

For example, how is it that the signal intensity is a measurement of the "magnetic field integral"? How do we know that the peak in the oscillation amplitude is the 0° phase point (and likewise that the first zero crossing is the 90° phase point)? I think that it would be good to help the reader out here with a concise explanation of the process that doesn't require them to trawl through multiple references!

While we understand the request of the reviewer, the description of spin echoes in a ^3He beam, which is needed to understand why a maximum occurs at a specific magnetic field integral is complex and we think it is beyond the scope of this paper. This is why we referred readers, interested in understanding this level of detail, to existing literature.

10. P7: If I understand this next section correctly, the authors now describe two configurations whereby the beam is blocked by a "sample" located in two positions. The real and imaginary magnetic field integral data ($S(kx)$) is then collected at two magnetic field values of B1 corresponding to the 0° and 90° phase conditions. This data is then Fourier transformed into the density distribution function. In the case of no sample, this is the beam profile. Again, this section is really challenging to understand and should be restructured for clarity:

The reviewer understood the process correctly. We have made changes to clarify the description following the detailed remarks below.

a. The authors should provide a picture showing the A and B configurations.

Changes: We have added schematic diagrams of the A and B configurations in figures 3 and 4 which together with the schematic in new figure 2 should make it clear what was measured.

b. P7 L8: Why is the encoding device only positioned through part of the beam?

The motivation here was to create characteristic non-uniform profiles which could then allow a more informative comparison between the profiles reconstructed from magnetic encoding and reference measurements performed by monitoring the pressure while scanning the wire.

Changes on page 7: To image a beam with a non-uniform profile, we positioned the encoding device such that the hollow tube the beam can pass through partially overlaps the position of the beam in space (configuration A, illustrated schematically as an inset in figure 3b).

Why does moving the encoding device lead to an apparent shift of the beam profile?

Moving the encoder changes the position of the overlap region through which the beam passes through the device. The schematics we added as insets in figure 3 should hopefully make this clear.

Changes: schematics of configurations A and B were added to figures 3 and 4.

What is the actual beam profile – is it the sum of the two profiles shown in Fig 3a?

Yes, more or less. Note that the beam diameter is about 2 mm and the hollow tube in the encoder is slightly smaller so a full profile of the uninterrupted beam cannot be obtained in one measurement.

Are the profiles shown in Fig 3a the convolution of the actual beam profile with the response function of the encoding device?

Yes, the red and magenta markers are exactly that, where the width of the “response function”, is dominated by the Fourier reconstruction resolution conditions described in the manuscript. We added text to clarify this.

Changes: Text (page 8) changed to “The red triangular markers in figure 3d present the magnitude of the Fourier transform of $S(k_x)$ and should correspond to the 1d spatial profile $\rho_{1d}(x)$, convoluted with the resolution function of the reconstruction”.

c. It would be helpful to have a figure that shows the workflow in getting the final image from the data (i.e. Figure 3b to Fig 4c and Fig 3a to Fig 4b).

Changes; Flowcharts have been added to the supplementary information (S6) to clarify the steps for 1d and 2d imaging.

d. It would be helpful to label the two configurations in Fig 3a.

Changes: The figures have been rearranged (see below) and the configurations are labelled.

e. Just a minor point, why label config A on the right and config B on the left in fig 4, and the reverse in Fig 3b and 3c? Its just confusing and unhelpful. It would make more sense for the labels to read A to B left to right in both cases.

Changes: We agree and have changed the labelling accordingly.

f. In my view, it would be better to have Fig 3b and Fig 3c in the same figure as profiles shown in Fig 4a – then the reader can see how one flows into the other.

Changes: We have changed the layout of the figures to make them clearer.

11. P8: There is no discussion of Fig 5b and Fig 5c in the text?

Figures 5b and 5c were referred to in page 13 but it was easy to miss that. In the revised manuscript they are now figures 4d and 4e (due to new figure layout). While they are only discussed towards the end of the manuscript we thought the best choice was to combine them with the other panels of figure 4 as they also show simulated 1d scans. Adding them as a separate figure seems a bit wasteful in terms of space.

12. P12-13: The discussion regarding the potential imaging time saving with increasing resolution is interesting. However, it is not clear in this comparison as to whether there is a difference in the signal to noise between the pinhole approach and the magnetically encoded approach. For example, what are the relative signal strengths between the two approaches? How would we actually compare between the two?

Indeed the discussion focusses on the scaling of the SNR/acquisition time with resolution rather than absolute values for SNR. Unfortunately until a dedicated magnetic encoding microscope is designed and built it would be very difficult to do so. We have however added several new

paragraphs in the discussion part of the paper (pages 12-14) where we discuss the prospects of magnetic encoding imaging and provide quantitative assessments as best as we can.

Changes: Discussion added in pages 12-14 discussing the prospects of magnetic encoding.

In addition, the following minor revisions should be addressed:

1. P6 L5: Provide a reference for the statement: "typically a FWHM of a few percent in supersonic beams"

Changes: Reference added.

2. Could the authors have a consistent x-axis distance scale for all of the figures.

In figure 3c we chose to show the full FOV acquired in the scan, as the lack of intensity outside the beam region is one of the tests showing the technique works. In figure 4 it seemed important to us to magnify the axes to the region of space where the image is, to allow the reader to see the fine details such as the change in the wire position and the comparison with the simulated data.

3. Could the authors make all of the axis labels and titles much larger!

Changes: The font size was increased.

4. P13 L20/21: Correct "loses" to "losses".

Changes: Text was changed, typo removed.

Reviewer #2 (Remarks to the Author):

Lowe et al. present the fascinating idea of performing MRI-style imaging experiments of atom-surface scattering to provide a new modality for surface microscopy. The work builds on ^3He spin-echo experiments that are now well established and provide unique insights into surface dynamics. The methodology is sound and the experiments have the potential to be of interest to the readership of Nature Comms. That said, I think that the work described is suited to a more specialist journal.

We would like to thank the reviewer for their report. We have made some significant changes to the manuscript following the remarks from both reviewers, we believe the manuscript in its new form is more accessible to a non-specialist readership. Red text is used below to mark the reviewers comments, whereas our answers and specific changes to the manuscript appear as black and blue text correspondingly .

Advances in microscopy normally relate to improvements in resolution or contrast. Even new techniques need to demonstrate or promise an advance in one of these aspects. Here, the spatial resolution is 50 microns, which is not competitive for neutral atom microscopy.

The manuscript presents a completely new methodology for spatial imaging in neutral beam experiments and presents proof of principle measurements. The only importance of the image resolution of the proof of principle experiments, is for quantitatively comparing the results to the reference measurements and to calculated profiles. These quantitative comparisons provide confidence in the derivations presented in the manuscript and are an essential step for further developing this technique and in particular for designing dedicated magnetic encoding microscopes in the future with competitive resolutions.

We would also like to stress that previous conceptual breakthroughs for achieving spatial resolution with neutral atoms which used mirrors or Fresnel plates to focus atoms (Nature 390, 244, (1997) & Phys. Rev. Lett. 67, 3231 (1991)), produced focussed spots of a few hundred microns in diameter. Nevertheless, these early works inspired the development of competitive microscopes decades later, which is what we hope this work will achieve as well.

Changes: To make it clear that the resolution only serves for quantitative comparisons with calculations, rather than represents what can be achieved with magnetic encoding we removed the reference to the resolution value in the abstract.

Unfortunately, there is no clear projection for what resolution might be obtained with the new technique. (Is one limiting factor, as in MRI, the maximum B fields and gradients that can be achieved?)

We accept this criticism, and we have now added several paragraphs discussion to project what might be possible with magnetic imaging in terms of resolution. These paragraphs also include a detailed discussion of how and what could be obtained from increasing the gradient field strength and what is likely to be the limiting factor on this.

Changes: Discussion was added in pages 12-14.

Although contrast is mentioned in the introduction, it is not described in detail, so it is unclear what the new technique might teach us about the atom-surface interaction or what new samples could be imaged. The promise of this new technique is thus unclear.

We agree that the contrast discussion in the previous version of the manuscript was brief and needed expanding. We have changed the text which now discusses the contrast mechanisms and sensitivity to nanoscale topography observed for ^4He in SHeM, and explains that these

mechanisms and sensitivity should be directly applicable to magnetic encoding imaging with ^3He . We also mention a potential new contrast mechanism related to surface magnetism, but stress that this remains hypothetical until proven by experiments.

Changes: New paragraph in page 12

“While the lateral resolution of existing SHeM setups is on the order of micrometres, the very short deBroglie wavelength of the particles and the fact they interact with the outermost electrons of the surface, means that SHeM has been successfully used to identify nano-metre topographic features which are orders of magnitude smaller than its lateral resolution, including differentiating the shape and chemical identity of single layers of adsorbed particles and even identifying local atomic-scale crystalline order seen in diffraction imaging mode.^{2,29,30} Since the interaction of ^3He and ^4He atoms with surfaces is very similar, we expect all the contrast mechanisms and topographic resolution observed in SHeM to apply to magnetic encoding imaging. A new contrast mechanism might become available with magnetic encoding imaging if the surface itself is magnetic and alters the spin state of the beam particles during scattering, thus relating the image of the surface to local magnetic properties. To the best of our knowledge nuclear spin flips have never been observed in ^3He scattering from non-magnetic surfaces, and until measurements showing spin-flips from magnetic surfaces become available, this potential contrast remains hypothetical.

I am not sure that the manuscript will be accessible to readers outside the (small) atom scattering community. This isn't essential for publication in Nature Comms but I think that it would be sad for the wider microscopy community not to appreciate the significance of the work.

We agree with the referee that it would be sad to not expose the wider microscopy community to this work. We have made significant changes to the text and to the figures to make it more accessible, we have also added new figures (fig2 and fig5) and animation videos (SV1 and SV2).

For example, the apparatus description needs a schematic.

We agree with the reviewer on this point. A schematic was added in new figure 2. The detailed description of magnetic encoding device (previous figure 2) has been moved to the supplementary information (S4) as it contains details which are not essential for understanding the main points of the manuscript.

Changes: Schematic added.

The discussion surrounding figure 6 is hard to follow.

We accepted this criticism, also mentioned by reviewer 1. We have changed the text and figures to convey in a more accessible way, how the scaling of the imaging (SNR) with the resolution of the image depends on the nature of the sample.

Changes: paragraphs in pages in pages 9-11 were completely rewritten, figure 5 was added and figure 6 was modified.

The description of the signal levels scaling with the '4th power of the pin hole dimension' will be unfamiliar to microscopists who are more used to descriptions of beam brightness and optically-limiting apertures.

We accept that this is a technical detail which most readers will not know about. However, explaining the strong dependence between flux and resolution in pin-hole microscopes is complicated and beyond the scope of this paper, and unfortunately readers interested in this will

have to read the referenced paper. We have however slightly reworded that sentence in a way which hopefully makes the comparison with magnetic imaging more understandable.

Changes on page 9:

“In an optimally designed pin-hole microscope, reducing the size of the beam spot on the sample and correspondingly the pixel size by a factor of N , will reduce the flux by N^4 as shown by Bergin et al.²⁸ On the other hand, magnetic encoding experiments do not require multiple microscopic apertures or microscopic beam sources for their resolution, and increasing the resolution means the number of atoms contributing to a pixel will at the worst case (for a sample with a relatively flat/smooth structure, as will be discussed further below) reduce linearly with each dimension, i.e. quadratically in 2d imaging. “

There is a comment in the final sections relating to the detector efficiency that will be unclear to most readers.

We have reworded this paragraph, hopefully the new text is clearer.

Change on page 14: “On the other hand, the beam flux at the sample, the detection efficiency, the reflectivity of the surface (in reflection mode) and how strong the contrast is, could all potentially become limiting factors. Before a dedicated magnetic encoding microscope is designed, built and tested on various samples it is impossible to quantify all of these factors. We note however that in our proof of principle experiments we used a commercial mass spectrometer which is approximately 5 orders less sensitive than state-of-the-art large ionisation detectors³³. We also used a large (2mm) beam instead of a brighter smaller beam source, which would be better suited for a microscopy experiment²⁸. Consequently, there is room for huge improvements by implementing already established technological improvements, making magnetic imaging a promising method for imaging with neutral beams with resolutions comparable and potentially even better than those available with current SHeM.”

How atoms scatter and depolarise during scattering, and if they still contribute to the detected signal isn't obvious.

Nuclear spin flips due to the scattering process have never been observed in the many helium-spin-echo studies which have been performed, hence depolarisation of the nuclear spin is not expected when performing imaging with these beams. A potential exception might be when scattering from magnetic surfaces where there is currently very limited data. If this happens it would open new contrast mechanisms as mentioned earlier.

Changes in manuscript (page 12): “To the best of our knowledge nuclear spin flips have never been observed in ³He scattering from non-magnetic surfaces, and until measurements showing spin-flips from magnetic surfaces become available, this potential contrast remains hypothetical.”

There are many such minor issues throughout the manuscript. Overall, I think that this is lovely work and a substantial technical achievement but not yet suitable for this journal. The promise of the new technique needs to be explained in greater detail and the description needs to be more accessible.

The changes we made to the manuscript include a much more significant discussion of the promise of the technique, the text was made clearer with some explanations given in the main text whereas other more technical points can be found in the supplementary information instead. The figures have been revised for clarity, schematics and block diagrams have been added to explain both the simplified imaging principle and the more complex experimental setup used for the proof of principle experiments, and finally two animation videos have been added which provide a graphical description of how the experiment works. We believe the result is more accessible and thank the two reviewers for their remarks which have driven these improvements.

REVIEWERS' COMMENTS

Reviewer #1 (Remarks to the Author):

The authors have made substantial changes to the manuscript that have improved considerably the readability and clarity of the work. The addition of the new videos in the supplementary section is a particular improvement and the authors are to be commended on the effort that they have put in.

All of the referee's concerns and comments have been addressed and the paper should be published as is.

Reviewer #2 (Remarks to the Author):

The authors have made worthwhile efforts to improve the clarity of the manuscript. This has been partially successful. The description is still technically dense and I find it hard to follow. I recommend passing the manuscript to a colleague in the MRI or microscopy communities to help 'translate' the molecular beam scattering concepts and to move complications to a supplementary section. It's still not clear what the new microscope could do that can't be done with neutral atom microscopy already. It's not clear how changes in scattering angle (not just intensity) with reflection from a sample could complicate the encoding process. I also wonder if the polariser assembly could be used as a simple optical condenser to improve the intensity of the helium beam at the sample plane.

All of these comments mean that I still admire the work and still think it of interest to the molecular beam community, which is enough for publication. I just think that the wider microscopy community will struggle to see the significance of the work.

Response to reviewers

Reviewer # 1: No comments to address.

Reviewer # 2: We understand that some technical points will still be difficult to follow. Given the restrictions on the length of the manuscript, we believe the current version of the text provides a good compromise for providing an understanding of the basic imaging scheme, whilst using references to cover technical points related to general concepts of atomic/molecular beams, which have been described elsewhere in the literature.

Regarding the question about the effect of the scattering angle on the encoding process, this should not make a difference if the encoding is performed before the interaction.

We have added text to clarify this point in the manuscript and added the following sentence in page 4 “We would like to emphasise that while magnetic encoding in a transmission experiment can be performed before or after the interaction, for a scattering experiment, encoding before the interaction has the advantage of avoiding complications related to the angular spread of the scattered beam.”.

Regarding the question about improving the intensity of the helium beam at the sample plane through focusing at the polariser. We do indeed use the polariser as a focusing element converting the initially diverging beam into a parallel beam. Further focusing however would be problematic as we need the beam to pass through the encoder with parallel trajectories to obtain non-ambiguous position information.

We have made a small change to the text to emphasise that the beam is parallel after passing through the polariser . The revised sentence in page 5 now reads “A hexapole magnet followed by a dipole field is used to produce a parallel and polarised beam¹⁹. “.